# OmniSegmentor: A Flexible Multi-Modal Learning Framework for Semantic Segmentation

**Bo-Wen Yin**[1,2], **Jiao-Long Cao**[1,2], **Xuying Zhang**[1,2], **Yuming Chen**[1,2],
**Ming-Ming Cheng**[1,2], **Qibin Hou**[1,2][†]
[1]NKIARI, Shenzhen Futian
[2]VCIP, College of Computer Science, Nankai University
[†]Corresponding author
bowenyin@mail.nankai.edu.cn

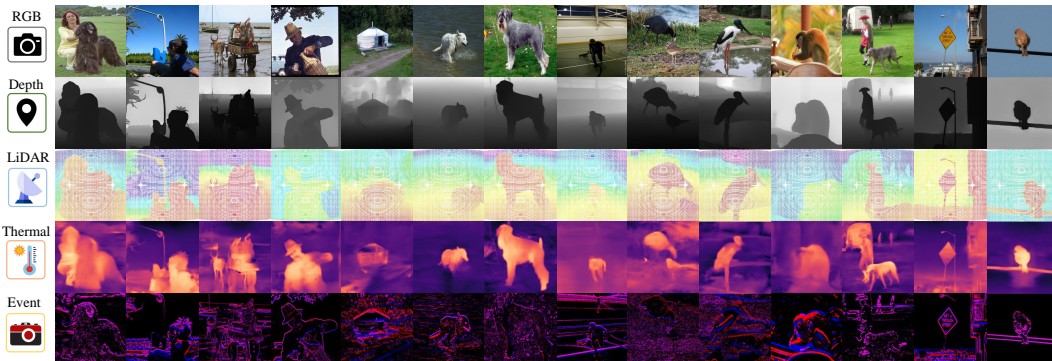

Figure 1: Visualizations of our assembled **ImageNeXt** dataset. Built upon ImageNet [44], a widely used large-scale RGB classification dataset, ImageNeXt is composed of five popular visual modalities for each sample, including RGB, Depth, LiDAR, Thermal, and Event.

## Abstract

Recent research on representation learning has proved the merits of multi-modal clues for robust semantic segmentation. Nevertheless, a flexible pretrain-and-finetune pipeline for multiple visual modalities remains unexplored. In this paper, we propose a novel multi-modal learning framework, termed **OmniSegmentor**. It has two key innovations: 1) Based on ImageNet, we assemble a large-scale dataset for multi-modal pretraining, called **ImageNeXt**, which contains five popular visual modalities; 2) We provide an efficient pretraining manner to endow the model with the capacity to encode different modality information in the ImageNeXt. For the first time, we introduce a universal multi-modal pretraining framework that consistently amplifies the model's perceptual capabilities across various scenarios, regardless of the arbitrary combination of the involved modalities. Remarkably, our OmniSegmentor achieves new state-of-the-art records on a wide range of multi-modal semantic segmentation datasets, including NYU Depthv2, EventScape, MFNet, DeLiVER, SUNRGBD, and KITTI-360. Data, model checkpoints, and source code will be made publicly available: https://github.com/VCIP-RGBD/DFormer.

## 1 Introduction

With the widespread use of modular sensors, multi-modal data for semantic segmentation is becoming more and more accessible. The knowledge perceived from multi-modal data can achieve more robust

39th Conference on Neural Information Processing Systems (NeurIPS 2025).

scene understanding, facilitating multi-modal learning research on a series of vision tasks. However, existing works [71, 72, 10, 58] usually employ RGB pretrained or randomly initialized weights to process different modalities, leading to mismatched encoding of the data [3]. Recent work DFormer [66] attempts to solve this issue using a new pretraining manner on the modality-specific scenes, *i.e.,* RGB-D. Considering the current trend of fusing more and more modalities [6, 72, 58], it would be of great interest to explore a flexible and efficient pretrain-and-finetune framework for multi-modal data, which was seldom researched before.

To construct such a flexible and efficient framework, several prominent problems should be considered. The foremost challenge is that multi-modal pretraining requires a large-scale dataset containing a variety of visual modalities. Although some existing datasets [18, 24, 34, 72, 48, 49] can partially satisfy this requirement, they either concentrate on a specific modality besides RGB images or have a limited scale of training samples, making them unsuitable for multi-modal pretraining. In addition, when the types of visual modalities increase, how to efficiently perform multi-modal pretraining and how to flexibly deploy the pretrained weights to downstream tasks with different types of visual modalities are still open questions.

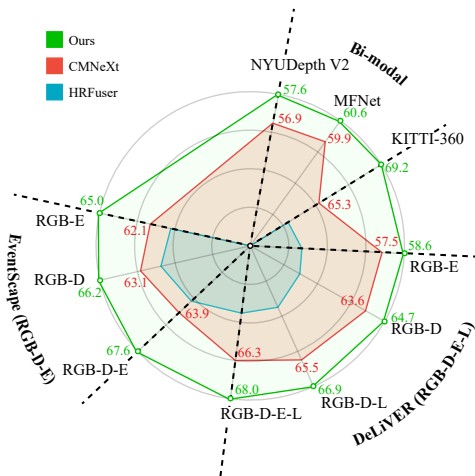

Figure 2: Performance comparisons between our OmniSegmentor and recent state-of-the-art methods (*e.g.*, HRFuser [6] and CMNeXt [72]) on various multi-modal semantic segmentation benchmarks. 'D, E, L, T' are abbreviations for depth, event, LiDAR, and the thermal modalities.

Taking the above analysis into account, in this paper, we attempt to construct a flexible and efficient pretrain-and-finetune framework for multi-modal semantic segmentation, named OmniSegmentor. Firstly, we need to address the issue of lacking large-scale multi-modal training data. Some methods [66, 3, 63] discover that synthetic data can compensate for this deficiency and improve the capacity of the model. For instance, DFormer [66] and DepthTrack [63] utilize synthetic depth data to perform multi-modal pretraining, thereby avoiding the mismatch between RGB pretrained models and RGB-D data and bringing significant improvement in RGB-D segmentation and tracking, respectively. Inspired by these works, we build a large-scale multi-modal dataset with synthetic data, called ImageNeXt, as shown in Fig. 1, to address the data issue and make the joint multi-modal pretraining feasible. This dataset is built upon ImageNet [44] and supplements each RGB image with four additional visual modalities, *i.e.,* depth, thermal, LiDAR, and event. We empirically found that the assembled dataset can help the model learn strong visual representations during pretraining.

Given the large-scale multi-modal data, the next challenge is to present an efficient method for multiple modalities. However, our experiments reveal that simultaneously pretraining a unified model on all the modalities not only imposes considerable computational burdens but also leads to optimization difficulties. To better accommodate the multi-modal data, we design a novel *pretrain-and-finetune pipeline* that can achieve efficient pretraining and flexible finetuning. To be specific, during pretraining, instead of simultaneously inputting all types of modality data at each iteration, we propose feeding the RGB data and a randomly selected other modality data into the model and conducting feature alignment. This simple strategy enables the model to efficiently absorb the patterns from different modality data, thus avoiding the mismatch problem between pretraining on RGB and finetuning on multi-modal data. Moreover, the training efficacy can be largely improved. For finetuning on downstream tasks, the weights corresponding to the supplementary modality in the pretrained model are used to initialize the weights for each supplementary modality. This approach allows each modality to be processed separately within each building block, thereby providing diverse informative features from different types of modality data for semantic segmentation. By adding a lightweight decoder head to the top of the ImageNeXt pretrained model, OmniSegmentor can generate high-quality predictions for different multi-modal segmentation tasks.

To the best of our knowledge, we are the first to construct a flexible pretrain-and-finetune pipeline for semantic segmentation with increasing visual modalities, *i.e.,* OmniSegmentor, composed of the ImageNeXt, well-designed pretraining and finetuning method. Extensive experiment results

demonstrate the effectiveness of OmniSegmentor on the benchmarks of a wide range of multi-modal semantic segmentation tasks, including NYU Depthv2 [48], EventScape [18], MFNet [24], DeLiVER [72], SUNRGBD [49], and KITTI-360 [34]. As shown in Fig. 2, our OmniSegmentor achieves new state-of-the-art records across all settings on all benchmarks. We hope that this work will provide new insights for multi-modal representation learning and set new baselines for multi-modal semantic segmentation.

## 2 Related Work

### 2.1 Multi-Modal Semantic Segmentation

Recently, significant advancements in semantic segmentation have been made as the rise of deep learning technologies, typified by CNNs [38, 26, 27, 33] and Transformers [39, 54, 15, 5]. However, most methods still struggle to cope with real-world scenes, as they only focus on processing RGB images [74, 51, 23, 36, 65], which lack sufficient information from other visual modalities, like LiDAR and depth. Multi-modal semantic segmentation has been explored by harvesting complementary information from supplementary modalities, such as depth [71, 9], thermal [73, 61, 47], LiDAR [64], and event [2, 70]. A series of methods are proposed to utilize the characteristics within other modalities for a more robust semantic segmentation. CMX [71] addresses multi-modal segmentation through multi-level cross-modal interactions, including channel and token exchanges. CMNeXt [72] introduces a universal multi-modal semantic segmentation framework with arbitrary modal complements. However, most existing relevant methods [58, 71, 72] employ RGB pretrained or randomly initialized weights to process the supplementary modalities, which may not fully extract the specific characteristics of each modality. To address this issue, DFormer [66] proposes to pretrain the encoder with RGB-D data to better leverage depth cues and alleviate the mismatch problem between pretraining and finetuning. Its significant improvement in both efficiency and effectiveness also emphasizes the importance of solving the mismatched encoding. However, DFormer is modality-specific (RGB-D) and is difficult to be applied to other modalities. Beyond the aforementioned works, we aim to provide a flexible and efficient pretrain-and-finetune framework that can efficiently perform multi-modal pretraining and flexibly deploy the pretrained weights to various downstream tasks.

### 2.2 Multi-Modal Representation Learning

Multi-modal representation learning endows models with the capacity to establish the relations among the specific information from multiple signal sources. The learned transferable representations can yield remarkable performance across various downstream tasks, as demonstrated in previous works [43, 67, 32]. Existing multi-modal learning methods encompass a large number of modalities, including image-text [8, 11, 37, 43, 75, 60], text-video [1], image-depth [20, 3], and image-text-audio [77, 21], etc. Structurally, these methods can be categorized into two types. The first type of method adopts separate encoders. They exploit multiple encoders to independently project the inputs of different modalities into a common space and minimize the distance between/among them or perform feature fusion. For instance, CLIP [43] employs two individual encoders to encode the image-text pairs and align them via contrastive learning. The second type of method adopts unified encoders to encode different modalities individually or multiple modalities jointly. Typically, Omnivore [20] and Meta-transformer [77] are able to process different modalities separately, while DFormer [66] and MultiMAE [3] can simultaneously deal with two visual modalities, *i.e.,* RGB and depth. However, the former cannot establish connections between different modalities, and the latter is limited to specific kinds of multi-modal data. An important reason that limits the development of multi-modal representation learning is the lack of a large-scale multi-modal dataset. For example, multi-modal datasets SUNRGBD [49] contains 10,325 RGB-D data, and KITTI-360 [34] has 61,280 RGB-L data, which are relatively small-scale and limited to specific multi-modal data, *i.e.,* RGB-L or RGB-D. Taking the above analysis into account, in this paper, we provide a large-scale dataset and a novel pretrain-and-finetune framework, enabling supervised pretraining on five types of modal data and flexible finetuning on downstream tasks.

# 3 ImageNeXt Dataset

Building upon the ImageNet dataset, the assembled ImageNeXt is a large-scale dataset for multi-modal representation learning. To the best of our knowledge, it is the first attempt to cover as many popular visual modalities as possible, including RGB, depth, thermal, LiDAR, and event. Unless otherwise specified, the ImageNet dataset in this paper refers to the original ImageNet-1K [44]. As the ImageNet dataset, the sample numbers of the training set and the validation set of ImageNeXt are 1.2M and 50K, respectively. We will describe some details of each visual modality data.

**RGB.** RGB images are the foundational visual modality in computer vision research. It contains information about objects' color, texture, shape, surroundings, etc. The RGB images in our ImageNeXt come from the ImageNet dataset [44], which is one of the most popular large-scale image datasets so far.

**Depth.** Depth maps provide 3D geometry information about range, position, and object contours. Combining RGB and depth enhances the ability to distinguish objects with similar colors and textures, especially when they occupy different spatial locations [71, 72]. Following DFormer [66], we employ a popular depth estimation method, *i.e.,* Omnidata [16], to produce depth maps for all the images in our ImageNeXt.

**Event.** Event data offers numerous advantages, including a high dynamic range, excellent temporal resolution, and immunity to motion blur. These qualities are crucial in dynamic scenarios with motion-related information, such as driving and flying scenes. The N-ImageNet [29] dataset acquires event data from an event camera that observes monitor-displayed images from ImageNet. We follow this work and employ the samples in N-ImageNet as the event data for the ImageNeXt.

**LiDAR.** LiDAR cameras can furnish dependable and precise spatial-depth information about the physical environment. Following the recent methods like CMX [71] and DeLiVER [72], we adopt the widely-used pseudo-LiDAR generation method [56] to generate the LiDAR data based on our synthetic depth maps of ImageNet. To maintain consistency between the LiDAR data and the RGB images in terms of representation, we adhere to the approach used in [83], which involves transforming LiDAR data into a format resembling a range-view image.

**Thermal.** The thermal sensor can detect temperature differences on the surface of objects, making it very suitable for finding thermally concealed objects or detecting temperature anomalies. It does not rely on visible light, but rather on the infrared radiation emitted by objects. According to our investigation, there is no method for thermal image estimation. Thus, we train a thermal estimation model, which imitates the depth estimation method adabins [4], on four RGB-T datasets VT821 [55], VT1000 [52], VT5000 [53], and FLIR [22]. Then we use it to generate the thermal data.

# 4 OmniSegmentor

## 4.1 Efficient Multi-Modal Pretraining

Multi-modal pretraining needs to align different modal features and build interaction among them, making it challenging to optimize and time-consuming. Existing works [58, 71, 28] mostly attempt to finetune the RGB pretrained backbone for the multi-modal scenes, as shown on the left of Fig. 3. However, the pretrained backbone for downstream task finetuning is often trained on RGB images, which is inconsistent with the multi-modal input data during finetuning. This may cause representation distribution shifts in that the multi-modal data is not considered during pretraining, and the RGB pretrained backbone may not effectively extract the special information within the supplementary modalities. We aim to explore a multi-modal pretraining manner to alleviate this issue by leveraging the proposed ImageNeXt dataset.

Given the ImageNeXt dataset where each sample has five modalities and a classification label, a straightforward way to implement multi-modal pretraining is to perform the classification optimization on all modalities simultaneously, as shown in the middle part of Fig. 3. This paradigm is also adopted in [76, 59], which uses modality-specific encoders to process multi-modal images. Under this setting, each visual modality needs to be encoded independently, and the interaction will be performed between the RGB images and each supplementary modality. However, such a pretraining method yields considerable computational cost, greatly decreasing the pretraining efficiency. More

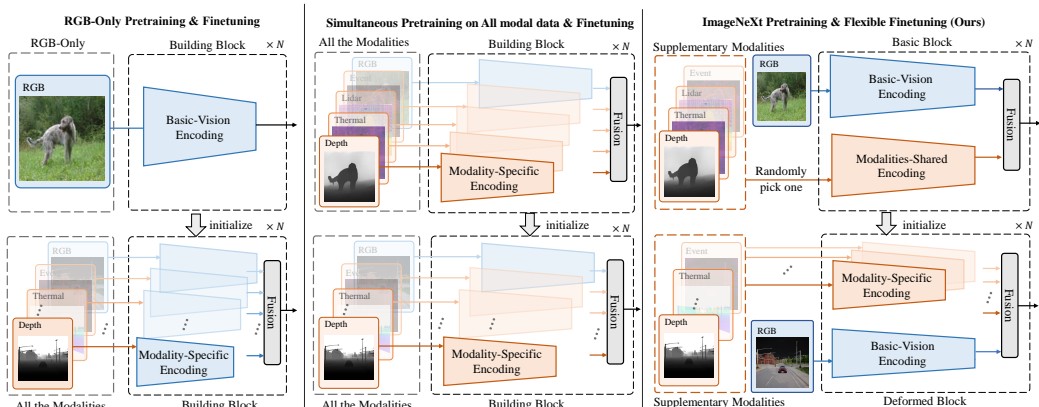

Figure 3: Illustration for different pretraining manners. The corresponding finetuning manner is also included. Left: RGB-only pretraining; Middle: simultaneous pretraining on all the modalities; Right: ImageNeXt pretraining of our OmniSegmentor. We omit the classification and segmentation heads for simplicity. The classification accuracy of the three manners is calculated with RGB input, all the modalities input, and the average on the RGB and each supplementary modality settings, respectively.

importantly, we observe that the above joint pretraining manner makes the optimization process difficult. As shown in Fig. 4, the training curve cannot converge well, and the Top-1 accuracy on ImageNet greatly decreases compared to pretraining on only RGB images. This issue also exists on the downstream multi-modal segmentation tasks as shown in Fig. 6.

To alleviate the above issue and meanwhile improve the pretraining efficiency, we propose an efficient multi-modal pretraining manner, called ImageNeXt pretraining. Instead of optimizing the model with all the modality data simultaneously, our method takes RGB images and a randomly selected supplementary modality as input. This paradigm is inspired by [20, 77, 21], which uses a single encoder to encode different modalities. Considering that processing RGB images is the primary factor affecting 2D semantic segmentation accuracy [3, 25, 66], we argue that assigning less computational load to supplementary modalities compared to RGB can lead to better performance and computation trade-offs. To implement this, we adopt an existing popular architecture, DFormer [66], which is originally designed for pretraining on RGB-D data. DFormer performs the simultaneous fusion of RGB and depth features from global and local views, as shown in the left part of Fig. 5. Meanwhile, it uses a base module to preserve the diverse appearance information within the RGB features. We find that such a block design can adapt well to our pretraining manner, though it is originally designed for RGB-D data.

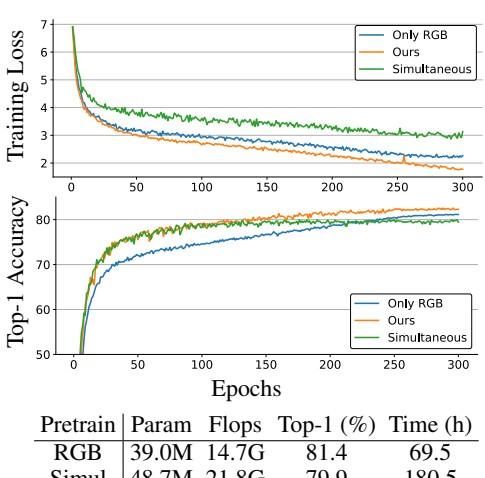

| Pretrain | Param | Flops | Top-1 (%) | Time (h) |
|----------|-------|-------|-----------|----------|
| RGB | 39.0M | 14.7G | 81.4 | 69.5 |
| Simul | 48.7M | 21.8G | 79.9 | 180.5 |
| Ours | 39.0M | 14.7G | 83.0 | 78.9 |

Figure 4: Curves of different pretraining manners on ImageNeXt. 'RGB': pretraining on RGB and removing the modality fusion operation; 'Simul': simultaneously pretraining on all the modalities; 'Ours': our ImageNeXt pretraining.

Our pretraining strategy offers the following advantages. First, each supplementary modality participates in the pretraining process. This makes the interaction between the RGB images and all the supplementary modality data efficient and hence can avoid the negative influence of other modalities on the representations of RGB images as much as possible. We empirically found that this strategy also improves the multi-modal encoding efficacy during finetuning on downstream tasks. Besides, the training process can be largely sped up. Particularly, compared to pretraining with all modality data as input, our strategy brings 3.1% gains in Top-1 precision shows consistent improvements in various downstream tasks in Fig. 6.

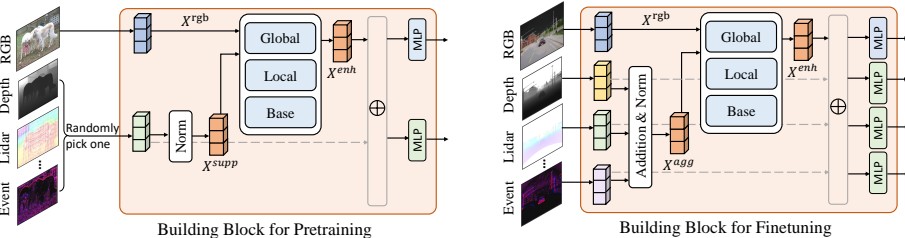

Figure 5: Building block of our OmniSegmentor. During pretraining, fusion modules aggregate the RGB features, and the features of the chosen modality, and the separate MLPs encode the features of different modalities. During finetuning, the sum of the features of supplementary modalities is fused with RGB features, and the features of different modalities are encoded separately by different MLPs.

## 4.2 Flexible Multi-Modal Finetuning

Given the pretrained model as described above, how to apply it to downstream tasks with multiple modality data as input is also important. The pretraining architecture in the left part of Fig. 5 is only suitable for processing data with an RGB image and a supplementary modality and is difficult to use for multiple supplementary modalities. Here, we present a flexible finetuning strategy and explain how to load the pretrained weights to initialize the model for downstream tasks.

During pretraining, the architecture adopts the modality-shared encoding for different supplementary modalities to efficiently absorb the patterns. Differently, during finetuning, we need to utilize all the provided modalities to perform robust semantic segmentation. In this situation, modality-specific encoding can better extract the unique characteristics within each supplementary modality, enabling the model to focus on different perspectives of the given scene. To achieve this, we use separate stem layers and MLPs for different modalities to implement modality-specific encoding, and the resulting features for different supplementary modalities are aggregated and then utilized to enhance RGB features, as shown in the right part of Fig. 5. Compared to the pretraining process, the model for finetuning has extra stem layers and MLPs to extract the characteristics within different supplementary modalities. The extra stem layers and MLPs are initialized by the pretrained stem layer and MLPs for the supplementary modality, while the other modules directly load the pretrained weights. We will describe the pipeline of OmniSegmentor for the finetuning in the following.

Given the RGB image and supplementary modalities, we first use different stem layers to separately process the input modalities. Then, the resulting features of different modalities are fed into the hierarchical encoder to encode multi-scale features. In each block, we first adopt an addition operation followed by a layer normalization to aggregate the information of all the supplementary modalities, and the aggregated feature is denoted as $X^{agg}$. The number of supplementary modalities can be arbitrary. Here, we empirically found that a more sophisticated fusion module will not bring further improvement compared to the above simple fusion operation, and the details are shown in Tab. 3. Then we fuse the RGB feature $X^{rgb}$ and the aggregated feature $X^{agg}$ to generate the enhanced feature $X^{enh}$ as follows as DFormer. After the fusion of modality clues, we add the $X^{enh}$ to the features of each modality as the output for each modality. The resulting features for each modality are processed by separate MLPs and then sent to the next block. For the decoder during finetuning, following DFormer, we adopt the Ham head [19] as the decoder. As a result, the OmniSegmentor is able to serve as an encoder for multi-modal segmentation tasks that input different modalities.

## 5 Experiments

### 5.1 Experiment Setup

To validate the effectiveness of our OmniSegmentor, we conduct extensive experiments on six popular multi-modal segmentation datasets, including NYU Depthv2 [48], SUNRGBD [49], MFNet [24], KITTI-360 [34], EventScape [18], and DeLiVER [72]. The experiments are conducted on NVIDIA A40 GPUs. The models are optimized using the cross-entropy loss function and the AdamW [30] method, where the learning rate is initialized to 6e-5 and scheduled by the poly strategy. The images are augmented by random resize with a ratio of 0.5 to 1.75, random horizontal flipping, and random

Table 1: Results on multimodal semantic segmentation datasets. 'D, E, L, T' are abbreviations for depth, event, LiDAR and thermal modalities, respectively. Follwoing [71, 72, 66], we adopt multi-scale inference in (a), (b), and single-scale inference in (c)-(f).

(a) NYUDepth V2 (RGB-D) [48].

| Method | Backbone | mIoU (%) |
|---|---|---|
| 3DGNN [42] | VGG-16 | 43.1 |
| CFN [35] | RefineNet-152 | 47.7 |
| ACNet [28] | ResNet-50 | 48.3 |
| Omnivore [20] | Swin-T | 49.7 |
| RDF-152 [40] | ResNet-152 | 50.1 |
| ESANet [45] | ResNet-34 | 50.3 |
| EMSANet [46] | ResNet-34 | 51.0 |
| SGNet [9] | ResNet-101 | 51.1 |
| DFormer [66] | DFormer-T | 51.1 |
| ShapeConv [7] | ResNext-101 | 51.3 |
| CEN [57] | ResNet-101 | 51.7 |
| NANet [69] | ResNet-101 | 52.3 |
| SA-Gate [10] | ResNet-101 | 52.4 |
| CEN [57] | ResNet-152 | 52.5 |
| Omnivore [20] | Swin-S | 52.7 |
| TokenFusion [58] | MiT-B2 | 53.3 |
| DFormer [66] | DFormer-S | 53.4 |
| FRNet [82] | ResNet-34 | 53.6 |
| PGDENet [81] | ResNet-34 | 53.7 |
| Omnivore [20] | Swin-B | 54.0 |
| TokenFusion [58] | MiT-B3 | 54.2 |
| CMX [71] | MiT-B2 | 54.4 |
| DFormer [66] | DFormer-B | 55.6 |
| MultiMAE [3] | ViT-B | 56.0 |
| CMX [71] | MiT-B4 | 56.3 |
| CMX [71] | MiT-B5 | 56.9 |
| CMNeXt [72] | MiT-B4 | 56.9 |
| DFormer [66] | DFormer-L | 57.2 |
| OmniSegmentor | ResNet-101 | 54.1 |
| OmniSegmentor | MiT-B2 | 56.8 |
| OmniSegmentor | DFormer-L | **57.6** |

(b) SUNRGBD (RGB-D) [49].

| Method | Backbone | mIoU (%) |
|---|---|---|
| CEN [57] | ResNet-101 | 50.2 |
| TokenFusion [58] | MiT-B2 | 50.3 |
| PGDENet [81] | ResNet-34 | 51.0 |
| TokenFusion [58] | MiT-B3 | 51.0 |
| CEN [57] | ResNet-152 | 51.1 |
| MultiMAE [3] | ViT-B | 51.1 |
| FRNet [82] | ResNet-34 | 51.8 |
| CMNeXt [72] | MiT-B4 | 51.9 |
| CMX [71] | MiT-B4 | 52.1 |
| CMX [71] | MiT-B5 | 52.4 |
| DFormer [66] | DFormer-L | 52.5 |
| OmniSegmentor | ResNet-101 | 51.7 |
| OmniSegmentor | MiT-B2 | 52.0 |
| OmniSegmentor | DFormer-L | **52.8** |

(c) MFNet (RGB-T) [24].

| Method | Backbone | mIoU (%) |
|---|---|---|
| ACNet [28] | ResNet-50 | 46.3 |
| PAP [78] | ResNet-18 | 50.5 |
| FuseSeg [50] | DenseNet-161 | 54.5 |
| ABMDRNe [73] | ResNet-18 | 54.8 |
| LASNet [31] | ResNet-152 | 54.9 |
| FEANet [13] | ResNet-152 | 55.3 |
| MFTNet [79] | ResNet-152 | 57.3 |
| GMNet [80] | ResNet-50 | 57.3 |
| DooDLeNet [17] | ResNet-101 | 57.3 |
| CMX [71] | MiT-B2 | 58.2 |
| DFormer [66] | DFormer-L | 59.6 |
| CMX [71] | MiT-B4 | 59.7 |
| CMNeXt [72] | MiT-B4 | 59.9 |
| OmniSegmentor | ResNet-101 | 59.0 |
| OmniSegmentor | MiT-B2 | 60.5 |
| OmniSegmentor | DFormer-L | **60.6** |

(d) KITTI-360 (RGB-L) [34].

| Method | Backbone | mIoU (%) |
|---|---|---|
| HRFuser [6] | HRFormer-T | 48.7 |
| PMF [83] | SalsaNext | 54.5 |
| TokenFusion [58] | MiT-B2 | 54.6 |
| TransFuser [41] | RegNetY | 56.6 |
| CMX [71] | MiT-B2 | 64.3 |
| CMNeXt [72] | MiT-B2 | 65.3 |
| DFormer [66] | DFormer-L | 66.3 |
| OmniSegmentor | MiT-B2 | 67.8 |
| OmniSegmentor | DFormer-L | **69.2** |

(e) EventScape (RGB-D-E) [18].

| Method | Modal | Backbone | mIoU |
|---|---|---|---|
| HRFuser [6] | RGB-E | HRFormer-T | 59.0 |
| CMX [71] | RGB-E | MiT-B2 | 61.9 |
| CMNeXt [72] | RGB-E | MiT-B2 | 62.1 |
| CMX [71] | RGB-E | MiT-B4 | 64.3 |
| OmniSegmentor | RGB-E | ResNet-101 | 61.5 |
| OmniSegmentor | RGB-E | MiT-B2 | 64.5 |
| OmniSegmentor | RGB-E | DFormer-L | **65.0** |
| HRFuser [6] | RGB-D | HRFormer-T | 59.9 |
| CMX [71] | RGB-D | MiT-B2 | 62.7 |
| CMNeXt [72] | RGB-D | MiT-B2 | 63.1 |
| CMX [71] | RGB-D | MiT-B4 | 64.8 |
| OmniSegmentor | RGB-D | ResNet-101 | 62.2 |
| OmniSegmentor | RGB-D | MiT-B2 | 64.9 |
| OmniSegmentor | RGB-D | DFormer-L | **66.2** |
| HRFuser [6] | RGB-D-E | HRFormer-T | 60.3 |
| CMX [71] | RGB-D-E | MiT-B2 | 63.0 |
| CMNeXt [72] | RGB-D-E | MiT-B2 | 63.9 |
| CMX [71] | RGB-D-E | MiT-B4 | 65.0 |
| OmniSegmentor | RGB-D-E | ResNet-101 | 62.8 |
| OmniSegmentor | RGB-D-E | MiT-B2 | 65.4 |
| OmniSegmentor | RGB-D-E | DFormer-L | **67.6** |

(f) DeLiVER (RGB-D-E-L) [72]. 'RGB-X' means the corresponding input modalities.

| Method | Backbone | RGB-E | RGB-D | RGB-L | RGB-D-E | RGB-E-L | RGB-D-L | RGB-D-E-L |
|---|---|---|---|---|---|---|---|---|
| HRFuser [6] | HRFormer-T [68] | 49.7 | 51.9 | 50.3 | 51.8 | 50.7 | 52.5 | 53.0 |
| CMX [71] | MiT-B2 [62] | 57.6 | 62.7 | 57.8 | 63.3 | 58.0 | 63.8 | 63.9 |
| CMNeXt [72] | MiT-B2 [62] | 57.5 | 63.6 | 58.0 | 64.4 | 58.9 | 65.5 | 66.3 |
| OmniSegmentor | MiT-B2 | 58.4 | **64.9** | 59.0 | 65.7 | 60.1 | 67.0 | 67.5 |
| OmniSegmentor | DFormer-L | **58.6** | 64.7 | **59.2** | **65.9** | **60.4** | **67.2** | **68.0** |

crop. More details, *e.g.*, pretraining settings, are in supplementary materials. Following DFormer [66], we adopts the light decoder head [19] by default. More experimental details are in the supplementary materials.

## 5.2 Comparison with Other Methods

Tab. 1 shows the comparisons of our OmniSegmentor against the recent state-of-the-art methods. In the following, we illustrate the results in two settings, *i.e.,* RGB with a single supplementary modality and RGB with multiple supplementary modalities.

**Single Supplementary Modality.** We first conduct experiments on four bi-modal segmentation datasets. As shown in Tabs. 1(a-d), our OmniSegmentor achieves new state-of-the-art records across all the four benchmarks. For the RGB-D segmentation benchmarks, our OmniSegmentor achieves 57.6% on NYUDepthV2 [48] and 52.6% on SUNRGBD [49], even better than the recent strong RGBD-specific pretrained methods, *i.e.,* DFormer [66], Omnivore [20] and MultiMAE [3]. For RGB-T segmentation on MFNet, our OmniSegmentor surpasses the recent SOTA CMNeXt (MiT-B4) by 0.7% mIoU. In addition, the performance of our OmniSegmentor achieves 69.2% on KITTI-360, which exceeds previous cutting-edge methods by a large margin (nearly +4%). Tab. 6 shows the parameters and Flops of our OmniSegmentor and the recent SOTA methods. As can be seen, our OmniSegmentor has lower computational cost compared to other methods but receives better results. Following [71, 72, 66], we adopt multi-scale inference in RGB-D semantic segmentation benchmarks, as shown in Tab. 1(a) and Tab. 1(b).

Table 2: Different modality settings within our ImageNeXt pretraining. Note that the pretraining duration is **100 epochs** in this experiment. We mark the significantly dropped performance in **bold**.

| Index | pretraining modalities RGB | Depth | Event | LiDAR | Thermal | NYU V2 RGB-D | MFNet RGB-T | KITTI RGB-L | EventScape RGB-E | EventScape RGB-D-E |
|---|---|---|---|---|---|---|---|---|---|---|
| 1 | ✓ | ✓ | ✓ | ✓ | ✓ | 54.3 | 57.6 | 64.6 | 61.8 | 63.8 |
| 2 | ✓ |   | ✓ | ✓ | ✓ | **52.2** | 57.5 | 64.6 | 61.6 | **61.9** |
| 3 | ✓ | ✓ |   | ✓ | ✓ | 54.2 | 57.6 | 64.5 | **60.5** | **62.9** |
| 4 | ✓ | ✓ | ✓ |   | ✓ | 54.3 | 57.7 | **61.2** | 61.9 | 63.7 |
| 5 | ✓ | ✓ | ✓ | ✓ |   | 54.3 | **56.4** | 64.8 | 62.1 | 63.8 |
| 6 | ✓ | ✓ | ✓ |   |   | 54.4 | **56.4** | **61.4** | 62.1 | 64.0 |
| 7 | ✓ | ✓ |   |   |   | 54.6 | **56.2** | **61.3** | **60.5** | **63.1** |
| 8 | ✓ |   |   |   |   | **50.9** | **55.6** | **60.1** | **58.7** | **59.7** |

**Multiple Supplementary Modalities.** Then, we carry out studies on two segmentation datasets with more visual modalities. As shown in Tab. 1(e), our OmniSegmentor surpasses all the other methods on the RGB-E and RGB-D of the EventScape dataset. Moreover, compared to CMNeXt, the advantage of OmniSegmentor is further enlarged from +0.7% (RGB-E), +1.4% (RGB-D), to +2.6% (RGB-D-E). Similarly, as shown in Tab. 1(f), our OmniSegmentor shows significant and consistent improvements across all seven settings of the input modalities on the DeLiVER dataset.

Furthermore, the ImageNeXt pretraining brings consistent improvements across all multi-modal segmentation benchmarks for different backbones, including ResNet-101, MiT-B2, and DFormer-L. For example, OmniSegmentor with the MiT-B2 backbone significantly exceeds other methods with the same backbone and even achieves better results than other methods with the MiT-B4 backbone.

## 5.3 Analysis on ImageNeXt Pretraining

**Pretraining manners.** To explain the necessity of our OmniSegmentor pretraining, we compare it with RGB-only pretraining and simultaneous multi-modal pretraining. Note that the modalities of the input data and the model structure are the same for the finetuning setting.

We adopt NYU Depthv2 [48], MFNet [24], KITTI-360 [34] and EventScape [18] to conduct this experiment. The results are shown in Fig. 6. The efficient pretraining of OmniSegmentor brings significant improvements in all benchmarks, e.g., 2.4% on MFNet and 5.1% KITTI-360. Specifically, on the EventScape benchmark, the improvement is increased from 3.3% to 4.2% on the RGB-D and RGB-D-E settings, illustrating that the improvement of the OmniSegmentor pretraining increases as the number of modalities increases. It demonstrates the multi-modal pretraining endows the model capacity to encode various modalities while the RGB-only pretrained model may not fully use other modalities. These experiments indicate that the multi-modal representation capacity learned at the pretraining stage of OmniSegmentor is crucial for robust semantic segmentation.

Figure 6: Comparison of the performance for different pretraining manners. 'RGB only': RGB-only pretraining; 'Simultaneous': simultaneously pretraining on all modalities; 'Ours': our ImageNeXt pretraining. Note that the input modalities when finetuning are the same for all the models.

**Ablation study on the pretraining modalities.** For more insights into our ImageNeXt pretraining, we take off different modalities during pretraining and then finetune it to verify whether the improvement is direct from the modal data used in our pretraining. We finetune the models that are pretrained on different modalities for multi-modal semantic segmentation tasks.

As shown in Tab. 2, first, from Row 2 to Row 5, we take off one of the supplementary modalities in our ImageNeXt pretraining. It is clear that the missing modality during pretraining leads to a

Table 3: Different fusion operations under different pretraining manners on EventScape (RGB-D-E). The pretraining duration of the model is 100 epochs.

| Fusion Operation | RGB-only | ImageNeXt |
|---|---|---|
| Simple fusion | 59.7 | 63.8 |
| SQ-Hub [72] | 61.0 | 63.7 |

Table 4: Effect of separate MLPs and shared MLP to encode supplementary modalities on the EventScape (RGB-D-E).

| Modality Encoding | Params | Flops | mIoU |
|---|---|---|---|
| Shared MLP | 39.0M | 68.9G | 66.7 |
| Separate MLPs | 41.9M | 68.9G | 67.6 |

significant performance drop in the settings that contain this modality. For example, in Row 3, the missing event modality during ImageNeXt pretraining leads to a significant performance drop in the RGB-E and RGB-D-E segmentation settings on the EventScape [18] dataset. This phenomenon demonstrates that the mismatched encoding may have an influence on the information extraction of the supplementary modalities.

From Row 6 to Row 8, we further take off more modalities. When adopting the ImageNeXt pretraining in RGB-D-E modalities, the performance of RGB-D-E semantic segmentation on EventScape is improved by 0.2%. Similarly, ImageNeXt pretraining on the RGB-D data brings 0.3% improvement on RGB-D segmentation on NYU DepthV2. Compared with the RGB-D pretraining settings, ImageNeXt brings significant improvements on all the other settings without sacrificing the performance of the RGB-D scenes. These results illustrate that a common weight for the supplementary modalities is sufficient to learn the information pattern within them during pretraining.

Moreover, we observe that ImageNeXt pretraining in RGB-D modalities also benefits segmentation tasks with RGB and other supplementary modalities, such as the effect in Row 7 and Row 8. Nevertheless, the improvement is relatively limited compared to the ImageNeXt pretraining with the corresponding modality. Similar phenomena also appear in Tab. 2. Even with part of the types of supplementary modality, the ImageNeXt pretraining can still help the model build the connection between the RGB and supplementary clues. For example, in Row 6 of Tab. 2, ImageNeXt pretraining without thermal also brings improvement on RGB-T segmentation compared to the RGB pretraining, *i.e.,* from 55.6 to 56.4 on MFNet. Meanwhile, ImageNeXt pretraining with thermal modality (Row 1) can further improve the RGB-T segmentation results, *i.e.,* from 56.4 to 57.6 on MFNet. These results illustrate that ImageNeXt pretraining with all the supplementary modalities is necessary.

**Discussion on the modalities fusion operation.** In Tab. 3, we compare the adopted simple fusion operation in our model with the self-query hub in the DeLiVER [72]. Under RGB-only pretrain, sophisticated fusion operation brings a significant improvement compared to the simple fusion operation, but it has no effect under the ImageNeXt pretrain. We hypothesize that the SQ-Hub may help alleviate the optimization difficulties in selecting information from different modalities from the random initialization weight, but our model can align the features of different modalities during the ImageNeXt pretraining thus alleviating the optimization difficulties in the finetuning.

**Separate MLPs or Shared MLP.** In our OmniSegmentor, we adopt separate MLPs to encode the unique characteristics within different supplementary modalities. In Tab. 4, we use the shared MLP to encode different supplementary modalities as a comparison to the separate ones. As can be seen, separate MLPs can bring significant improvement with a small increase in parameters. We hypothesize that specific parameters help extract the unique characteristics of each supplementary modality and achieve more robust segmentation results.

## 6 Conclusions and Future Work

In this paper, we propose a flexible framework for multi-modal segmentation, which is composed of the ImageNeXt dataset and the pretrain-and-finetune method. To the best of our knowledge, OmniSegmentor is the first framework to endow the model with the capacity to jointly encode more than three types of multi-modal data during pretraining. Benefiting from the modality selection mechanisms, OmniSegmentor can be applied for different multi-modal scenes, presenting robust segmentation across all the multi-modal scenes.

In the experimental part, we have conducted extensive experiments on various multi-modal segmentation benchmarks. However, these existing benchmarks are limited in scope, as they cover only a subset of the five major visual modalities, and some rely on synthetic data generated by simulation tools. In the future, we will attempt to gather more comprehensive multi-modal data from the real world and explore unsupervised/self-supervised pretraining manners to perform multi-modal learning.

**Acknowledgments** This work was partially funded by NSFC (No. 62495061, 62276145), National Key Research and Development Project of China (No. 2024YFE0100700), the Science and Technology Support Program of Tianjin, China (No. 23JCZDJC01050), Shenzhen Science and Technology Program (JCYJ20240813114237048), and the Fundamental Research Funds for the Central Universities (Nankai University) under Grant 070-63253220.

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

# A  Experiment Details

## A.1  RGBX benchmarks

To validate the effectiveness of our OmniSegmentor, we conduct extensive experiments on six popular multi-modal segmentation datasets. The statistics of the datasets and the corresponding training strategy of our model are shown in Tab. 5. We conduct extensive experiments with our OmniSegmentor on different multi-modal segmentation datasets and briefly introduce them in the following. NYU Depthv2 (RGB-D) [48] contains 1,449 RGB-D images with a size of 640×480, which is divided into 795 training and 654 test images with annotations for 40 categories. SUNRGBD (RGB-D) [49] includes 10,335 samples with 530×730 resolution. There are 37 semantic categories. Following [66], we randomly crop and resize the input to 480×480 during training. MFNet (RGB-T) [24] is a multi-spectral RGB-T image dataset, which has 1,569 images. 784/392/393 samples are used for training/validation/test, respectively, annotated in 8 classes at the resolution of 640×480. KITTI-360 [34] is a suburban driving dataset that contains 19 classes as same as the Cityscapes dataset [12]. EventScape [18] was originally designed for using RGB and event data to conduct depth estimation. It has pixel annotations for semantic segmentation as well. Following CMX [71], we select one frame from every 30 frames, obtaining 4,077/749 for training and evaluation to maintain data diversity from the original sequences generated by the CARLA simulator [14]. DeLiVER [72] is a large-scale multi-modal segmentation dataset, which is also generated by the CARLA simulator. This dataset contains 7,885 front-view samples divided into 3,983 / 2,005 / 1,897 for training / validation / test, respectively. each of which contains two types of annotations (*i.e.,* semantic and instance segmentation labels).

Table 5: Statistics of the used multimodal segmentation datasets and the corresponding training settings used in the proposed method.

| Datasets | NYU DepthV2 [48] | SUNRGBD [49] | MFNet [24] | KITTI-360 [34] | EventScape [18] | DeLiVER [72] |
|---|---|---|---|---|---|---|
| Modalities | RGB-D | RGB-D | RGB-T | RGB-L | RGB-D-E | RGB-D-E-L |
| Train/val/test split | 795 / 654 / - | 5285 / 5050 / - | 1568 / 392 / 393 | 49,004 / 12,276 / - | 4,077 / 749 / - | 3983 / 2005 / 1897 |
| Classes | 40 | 37 | 8 | 19 | 12 | 25 |
| Input size | 640×480 | 480×480 | 640×480 | 1408×376 | 512×256 | 1024×1024 |
| Batch size | 8 | 16 | 8 | 16 | 4 | 8 |
| Epochs | 500 | 300 | 500 | 40 | 100 | 200 |
| Base lr | 6e-5 | 8e-5 | 6e-5 | 6e-5 | 6e-5 | 6e-5 |
| Optimizer | AdamW | AdamW | AdamW | AdamW | AdamW | AdamW |
| Weight decay | 0.01 | 0.01 | 0.01 | 0.01 | 0.01 | 0.01 |
| Lr schedule | Linear decay | Linear decay | Linear decay | Linear decay | Linear decay | Linear decay |
| Stochastic depth | 0.35 | 0.20 | 0.40 | 0.30 | 0.20 | 0.30 |

## A.2  Experiment Setup

**Pretraining details.** The multi-modal features from the last stage are flattened along the spatial dimension and fed into the linear projection to obtain the category probabilities, which are used to calculate the classification loss, *i.e.,* the standard cross-entropy loss. To verify the universality and robstness of our methods on different architectures, we adopt DFormer-L [66], MiT-B2 [62], and ResNet-101 [26] as backbone and perform ImageNeXt pretraining with them. In the experiments, unless otherwise specified, the OmniSegmentor uses the DFormer-L backbone. ImageNeXt pretraining adopt the same hyperparameters as DFormer-L. Following the commonly used pretraining durations [38, 62, 23, 66], OmniSegmentor is pretrained for 300 epochs. We use AdamW [30] with learning rate 1e-3 and weight decay 5e-2 as our optimizer, and the batch size is set to 1024. We adopt the same data augmentation strategies as DFormer [66], *i.e.,* data augmentation strategies related to color, e.g., auto contrast, are only used for RGB images, while other common strategies are simultaneously performed on all the modalities, e.g., random rotation.

**Finetuning details.** The experiments are conducted on NVIDIA A40 GPUs. The models are optimized using the cross-entropy loss function and the AdamW [30] method, where the learning rate is scheduled by the poly strategy. The images are augmented by random resizing with a ratio of 0.5 to 1.75, random horizontal flipping, and random cropping. Tab. 5 presents the detailed training settings for different segmentation datasets. In Tab. 1, we compare our OmniSegmentor with other SOTA methods on all multi-modal benchmarks. The segmentation of RGB and multiple supplementary modalities is an emerging field. In EventScape (RGB-D-E) and DeLiVER (RGB-D-E-L), the results of some methods are missing. For example, CMNeXt [72] lacks the results on the

Table 6: Comparisons of our OmniSegmentor with other SOTA methods on the number of parameters and Flops. The calculation is based on the same code, and these methods are evaluated when keeping the inference settings in the corresponding papers. The inference time, *i.e.,* frames per second (FPS), is calculated on a single NVIDIA A40 GPU. When calculating FLOPs, the input size is set to 480×640.

| Methods | Ours | CMNeXt-B4 | CMNeXt-B2 | CMX-B5 | CMX-B4 | TokenFusion | MultiMAE | HRFuser | Omnivore |
|---------|------|-----------|-----------|--------|--------|-------------|----------|---------|----------|
| Params | 39.0M | 119.6M | 58.8M | 181.1M | 139.9M | 45.9M | 95.2M | 30.5M | 95.7M |
| FLOPs | 65.7G | 131.9G | 62.9G | 167.8G | 134.3G | 94.4G | 267.9G | 223.0G | 109.3G |
| FPS | 28.0 | 13.7 | 27.2 | 10.1 | 13.0 | 12.6 | 20.0 | 18.4 | 10.5 |

RGB-R-L setting. To make the comparison more comprehensive, we implement these methods on the missing settings based on their official code.

