# OpenReview forum: "OmniSegmentor: A Flexible Multi-Modal Learning Framework for Semantic Segmentation"
_NeurIPS.cc/2025/Conference — NeurIPS 2025 poster_

### Official Review · Reviewer_TR9e · 2025-06-17

**Clarity:** 4
**Significance:** 4
**Originality:** 3
**Rating:** 5
**Confidence:** 4

**Summary:**

OmniSegmentor: A Flexible Multi-Modal Learning Framework for Semantic Segmentation proposed ImageNeXt as assembled large-scale dataset for multi-modal training on various segmentation benchmarks including RGB, depth, event, LiDAR, and the thermal modalities. Based on an earlier work of DFormer, modality-specific encoding modules enable to train and evaluate the method on different datasets with different inputs. OmniSegmentor sets state-of-the-art results on NYU-D_v2, MFNet, KITTI-360, DeLiVER, and EventScape datasets.

**Questions:**

Is there any specific challenge in scaling the approach to more modalities or different domains?

**Ethical Concerns:**

["NO or VERY MINOR ethics concerns only"]

**Limitations:**

The limitations are noted in the text.

The real-world generalization and dependence on backbone choice could be explored in more depth.

**Paper Formatting Concerns:**

.

**Quality:**

4

**Strengths And Weaknesses:**

Strengths:

The paper proposes new multi-modal training benchmark that supports setting state-of-the-art results on various datasets.

The paper provides a user-friendly and flexible training framework for integrating multiple modalities, which can benefit real-world applications.

Strong empirical results are reported on several standard multi-modal segmentation benchmarks.

The release of data, code, and checkpoints will facilitate reproducibility and further research.

Weaknesses & Suggestions:

The approach primarily builds on existing models, with the main contribution being the training pipeline; architectural or methodological novelty is slightly limited.

Typo errors in the text, such as “Follwoing” [in Table 1], can be corrected.

---

> ### Author Rebuttal · Authors · 2025-07-30
>
> We thank the reviewer for the valuable feedback. We are happy to see your positive comments on experiments and method. We will release our data, code, and checkpoints once the paper is made public. We hope the following responses could solve your concerns.
>
> **1. About the Contribution.**
>
> Thank you for your thoughtful comment. We understand the concern regarding architectural or methodological novelty, and we appreciate the opportunity to clarify the primary focus of our work.
>
> Currently, there is no effective pretraining strategy capable of learning representations for such a diverse set of modalities. DFormer, for instance, was designed for two-modality (RGB-D) scenarios and cannot be directly applied to arbitrary multi-modal settings. As illustrated in Figure 4 of the main text, directly combining and optimizing all modalities simultaneously, in a manner similar to DFormer, introduces significant optimization challenges due to the complexity and heterogeneity of the data.
>
> To address this issue, we propose an efficient multi-modal pretraining strategy, termed ImageNeXt pretraining. Instead of optimizing the model with all modality data at once, our method takes RGB images and a randomly selected supplementary modality as input at each iteration. This design reduces optimization difficulty while maintaining the ability to learn transferable cross-modal representations.
>
> While our approach builds upon existing backbone models such as DFormer, our main contribution lies in designing a general and scalable training paradigm for multi-modal pretraining.
> Our unified pretraining pipeline enables the model to learn from multi-modal data and supports flexible fine-tuning under arbitrary modality settings. This strategy enhances the model’s adaptability and generalization in real-world multi-modal tasks.
>
> Although DFormer serves as a backbone in some experiments, we also demonstrate that the same pretraining paradigm can be effectively applied to other architectures such as ResNet-101, suggesting that our method is not limited to a particular design but offers broader applicability.
>
> We will revise the manuscript to better emphasize this motivation and the scope of our contribution. Thank you again for your valuable feedback, which has helped us improve the clarity of our work.
>
>
>
> **2. About Real-world Generalization and Backbone Choice.**
>
> Thank you for your valuable feedback. We agree that exploring both real-world generalization and the dependence on backbone choice is important for fully understanding the effectiveness of our approach.
>
> Regarding real-world generalization, our pretraining strategy leverages synthetic multi-modal data to equip the model with the capacity to handle diverse modality combinations. We then fine-tune the model on real-world downstream datasets, covering a wide range of practical scenarios—e.g., NYU DepthV2 and SUN RGB-D for indoor environments, and KITTI-360 and MFNet for outdoor scenes. As demonstrated in our results, the pretrained models consistently improve performance across these real-world benchmarks, indicating good transferability.
>
> We appreciate the suggestion and agree that further validation in real deployment scenarios would be beneficial. In future work, we plan to collect and evaluate on additional real-world multi-modal datasets to more thoroughly assess robustness in unseen conditions.
>
> In terms of backbone dependence, our method is designed to be generally applicable and not tied to a specific architecture. In the current version, we have already evaluated our training paradigm with multiple popular backbones, including DFormer, ResNet, and MiT (SegFormer), as shown in Table 1. Across these backbones, our method brings consistent performance improvements. We will further expand our ablation studies to include more backbones and explicitly highlight these results to better demonstrate the generality of our approach.
>
> We will conduct these additional experiments and incorporate them into the manuscript, which we believe will make the paper more convincing and solid. Thank you again for your helpful suggestions.
>
>
>
> **3. About Scaling to More Modalities and Domains.**
>
> Thank you for raising this important question.
> Our approach is explicitly designed to scale to additional modalities and domains without architectural or procedural changes. In the ImageNeXt pretraining paradigm, all supplementary modalities are treated equally, without requiring any modality-specific architectural customization. When new modalities are introduced, we can simply expand the modality pool and perform random sampling across modalities during pretraining. This ensures that the model is exposed to diverse cross-modal combinations and learns to handle new modalities in a unified manner.
>
> Furthermore, we find encouraging evidence that the representations learned through ImageNeXt pretraining generalize beyond the specific modalities involved in training. As shown in Table 9 of the supplementary material, ImageNeXt pretraining significantly boosts performance on RGB + Polarization semantic segmentation, even though the polarization modality was not present during pretraining. We hypothesize that the multi-modal fusion patterns and generalizable representations learned during pretraining enable effective transfer to new modality pairs, demonstrating strong scalability and domain generalization capabilities.
>
> We will clarify this point and include the corresponding analysis in the revised paper to better highlight the extensibility of our approach. Thank you again for the insightful question.
>
> Thank you for pointing out the typographical issues. We will carefully proofread and polish the writing to correct such errors and improve the overall quality of the paper.

---

> ### Comment · Area_Chair_X6ma · 2025-08-05
>
> Dear Reviewer,
>
> Thank you for your dedicated efforts in reviewing this paper. We are currently in the reviewer-author discussion phase, but we have not yet seen your engagement.
>
> This year's Responsible Reviewing initiative requires all reviewers to communicate with authors during this period, emphasizing that ghosting is not acceptable. We kindly ask that you reply and engage with the authors. Please note that participation in discussions with authors is mandatory before submitting "Mandatory Acknowledgement," as submitting it without any engagement is not permitted in this review cycle.
>
> Best,
> AC

---

### Official Review · Reviewer_Eebt · 2025-06-18

**Clarity:** 3
**Significance:** 3
**Originality:** 3
**Rating:** 4
**Confidence:** 4

**Summary:**

The paper presents a novel multi-modal dataset based on ImageNet, containing different synthetic visual modalities beside RGB. Together with the dataset, an efficient and effective pertaining strategy is shown, where and RGB Tower is augmented with a second encoder fed with a random additional modality during pretraining, reducing the mismatch across different visual modalities. The pretrained multimodal encoder is then fine-tuned for downstream tasks, potentially leveraging all modalities together.

**Questions:**

How does DFormer work? Which architecture does it leverage?

**Ethical Concerns:**

["NO or VERY MINOR ethics concerns only"]

**Final Justification:**

The dataset contribution may be relevant to the field, plus training strategy and method are pretty interesting.

**Limitations:**

Yes.

**Paper Formatting Concerns:**

None.

**Quality:**

3

**Strengths And Weaknesses:**

Strenghts:
- the paper is easy to follow;
- a new large scale multimodal dataset is presented;
- the new pretraining strategy is efficient and yields better results wrt naive pretraining with all modalities;
- the proposed model outperforms competitors across a wide set of benchmarks.

Weaknesses:
- being the paper largely based on DFormer, this model should be at least briefly described from an architectural and training point of view;
- the gain over competitors is very limited in certain experiments (e.g. Table 1 a and b);
- the use of synthetic data may hinder real world applications, although the effect seems to be limited when fine-tuning.

---

> ### Author Rebuttal · Authors · 2025-07-30
>
> Thanks for your effort for reviewing our paper and giving valuable suggestions. We are happy to see your positive comments on writing, experiments and method. We hope the following responses could solve your concerns.
>
> **1. Details about DFormer.**
>
> Thank you for your suggestion. We have briefly described DFormer's architecture and training strategy in the main text (lines 193–203), but we agree that a clearer exposition would improve the completeness of the paper.
>
> To clarify, DFormer is a unified RGB-D pretraining framework that learns transferable representations directly from image-depth pairs. Architecturally, it adopts a hierarchical encoder-decoder structure built on customized RGB-D blocks, which integrate RGB and depth features via two core modules: Global Awareness Attention (GAA) for global semantic fusion and Local Enhancement Attention (LEA) for local detail refinement. For pretraining, DFormer uses synthetic depth maps derived from ImageNet-1K and is trained with a classification objective. During finetuning, only RGB features are used by a lightweight decoder to perform segmentation tasks.
>
> While DFormer has demonstrated the effectiveness of multi-modal feature fusion in RGB-D pretraining, the core contribution of our work goes beyond validating multi-modal pretraining effectiveness. Our OmniSegmentor proposes a general and extensible multi-modal pretraining framework that addresses several critical limitations of prior work:
> (1) Modality Extensibility: Our framework supports arbitrary modality combinations (e.g., RGB-depth-thermal-event-LiDAR and RGB-depth-thermal), whereas DFormer is limited to RGB-D inputs.
> (2) Unified Pretraining Framework: Instead of designing modality-specific fusion modules, we propose a unified pretraining strategy that dynamically adapts to varying modality availability and combinations during training.
> (3) Dataset Innovation: We construct a large-scale synthetic dataset, ImageNeXt, which covers five modalities and provides a scalable and reproducible infrastructure for future multi-modal pretraining research.
>
> We will revise the paper to include a more explicit description of DFormer's architecture and training pipeline, along with a discussion highlighting the key distinctions and innovations introduced by our proposed approach. We appreciate your constructive feedback, which will help clarify our contribution and position it more precisely in relation to prior work.
>
>
> **2. About the Performance Gains in RGB-D Scenes.**
>
> Thank you for your valuable feedback.
> While the gains in Tables 1(a) and 1(b) may appear modest, it is important to note that the second-best method, DFormer, is specifically designed for RGB-D segmentation and also utilizes large-scale ImageNet (1.2M RGB images) with RGB-D pairs for pretraining. In contrast, our method adopts a more general and modality-agnostic design, incorporating additional modalities beyond RGB-D and enabling a unified pretraining strategy applicable to a wide range of multi-modal tasks.
>
> Despite this broader design, our model still achieves consistent improvements even on tasks where DFormer is highly specialized, highlighting the robustness of our approach. More significantly, our method demonstrates strong transferability across modalities and architectures, which we further validate on diverse downstream benchmarks.
>
> We will make this broader perspective more explicit in the revised version. Thank you again for your insightful comment.
>
>
> **3. Synthetic Data may Hinder Real World Applications.**
>
> Thank you for your insightful comment. In this paper, the synthetic multi-modal data is used for large-scale pretraining to equip the model with capability to process diverse multi-modal data. This is followed by fine-tuning on downstream datasets, which enables the model to adapt to different multi-modal scenarios.
>
> As shown in our experiments, this strategy leads to consistent improvements across multiple real-world benchmarks. For example, NYU DepthV2 and SUN-RGBD represent indoor scenes, while KITTI-360 and MFNet capture outdoor environments. These results demonstrate that our pretraining paradigm effectively benefit the real-world scene perception, despite being pretrained on synthetic data.
>
> We appreciate the reviewer’s concern and fully agree that exploring real data sources is a meaningful direction. In future work, we plan to incorporate more real-world multi-modal data and evaluate our method's robustness on previously unseen scenarios.

---

> > ### Comment · Reviewer_Eebt · 2025-08-06
> > **I confirm my review**
> >
> > Thank you for the nice rebuttal, I confirm my positive rating.

---

> > > ### Author Response · Authors · 2025-08-06
> > >
> > > We sincerely thank the reviewer for the positive evaluation of this paper. We will further improve our revised version based on the reviewer's comments.

---

> ### Comment · Area_Chair_X6ma · 2025-08-05
>
> Dear Reviewer,
>
> Thank you for your dedicated efforts in reviewing this paper. We are currently in the reviewer-author discussion phase, but we have not yet seen your engagement.
>
> This year's Responsible Reviewing initiative requires all reviewers to communicate with authors during this period, emphasizing that ghosting is not acceptable. We kindly ask that you reply and engage with the authors. Please note that participation in discussions with authors is mandatory before submitting "Mandatory Acknowledgement," as submitting it without any engagement is not permitted in this review cycle.
>
> Best,
> AC

---

### Official Review · Reviewer_HYgf · 2025-06-28

**Clarity:** 3
**Significance:** 3
**Originality:** 3
**Rating:** 4
**Confidence:** 4

**Summary:**

This paper presents OmniSegmentor, a multi-modal semantic segmentation framework trained on a large-scale synthetic dataset, ImageNeXt, covering five visual modalities. The model extends DFormer to support arbitrary modality combinations and achieves strong results on several benchmarks.
However, all non-RGB modalities are synthesized from RGB using public tools, limiting cross-modal diversity. The model introduces minimal architectural innovation, and performance gains from large-scale pretraining are marginal, raising concerns about the practical value and novelty of the work.

**Questions:**

Table2 doesn't seem to include all the modality combinations. Has the paper systematically explored all combinations of RGB with a single additional modality (e.g., RGB+LiDAR, RGB+Thermal, RGB+Event)?

**Ethical Concerns:**

["NO or VERY MINOR ethics concerns only"]

**Final Justification:**

The author solves my concern. Thus I keep my original score.

**Limitations:**

yes

**Paper Formatting Concerns:**

yes

**Quality:**

3

**Strengths And Weaknesses:**

## Strength
Assemble a large-scale dataset and introduce a framework that achieves good results on multimodal semantic segmentation datasets.

## Weaknesses

- Limited Dataset Contribution

ImageNeXt introduces multiple modalities, but all supplementary ones (depth, LiDAR, thermal, event) are *synthesized from RGB* using off-the-shelf tools like Omnidata and pseudo-LiDAR. This essentially reduces multimodal learning to **an advanced form of RGB data augmentation**, rather than capturing genuine cross-modal relationships from independent sensors. Since all modalities stem from the same RGB source, they exhibit strong inter-modal correlation and offer limited diversity. The dataset is easily reproducible using public methods, which diminishes its originality and contribution.

- Minimal Gains from Large-Scale Pretraining

As the authors mentioned, ImageNeXt is a large-scale dataset (1.2M samples) comprising five visual modalities, designed to be distinguishable from existing datasets. However, despite pretraining on three additional visual modalities (L/T/E) compared to its baseline (DFormer, which uses only RGB-D pairs), the improvements are marginal (+0.4% in Table 1.a and +0.3% in Table 1.b), raising concerns about the effectiveness and efficiency of large-scale synthetic data generation.

---

> ### Author Rebuttal · Authors · 2025-07-30
>
> Thanks for your time for reviewing our paper and giving some insightful suggestions. We hope the following responses could solve your concerns.
>
> **1.About the Dataset Contribution.**
> Thank you for your thoughtful comments regarding the nature of the supplemental modalities in our ImageNeXt dataset.
>
> While it is true that the additional modalities are synthesized from RGB using established tools, we emphasize that each modality retains its unique structural and statistical characteristics. Pretraining on these synthesized data enables the model to effectively learn and capture modality-specific representations.
> As shown in Table 2 of the main paper, pretraining on the synthesized ImageNeXt significantly improves performance on real-world downstream tasks, indicating that our synthetic modalities successfully capture modality-specific patterns essential for effective representation learning.
>
> Furthermore, the synthetic nature of our dataset is a deliberate design choice. It enables a scalable and reproducible paradigm for generating diverse multi-modal pretraining data, where multi-modal signals can be constructed directly from RGB images without requiring physical sensors. This allows the framework to be extended to construct even larger-scale pretraining data. We believe this aspect of ImageNeXt offers a practical and impactful contribution to the community, potentially serving as a foundation for future large-scale multi-modal dataset construction.
>
> We will highlight these clarifications more explicitly in the revised version. Thank you again for raising this important point.
>
>
> **2.About the Performance Gains on RGB-D benchmarks.**
>
> Thank you for the feedback. We would like to clarify the positioning and broader implications of our pretraining paradigm.
>
> First, it is worth noting that the second-best method on RGB-D benchmarks, DFormer, also leverages large-scale ImageNet (1.2M RGB samples) for pretraining, and it is specifically designed for RGB-D segmentation tasks. Its pretraining is conducted on RGB-D pairs and tailored to extract complementary features from depth and RGB inputs for that specific scenario.
>
> In contrast, our approach is designed to be more general and modality-agnostic. It incorporates additional modalities beyond RGB-D, and establishes a unified pretraining strategy applicable to a wider range of multi-modal tasks. Despite this generality, our model still shows improvements even on the RGB-D tasks where DFormer is highly specialized, as shown in Tables 1(a) and 1(b). We believe this underscores the robustness of our pretraining approach.
>
> More importantly, the value of our method lies not only in performance improvements on any single task, but in its transferability and broad applicability across modalities and architectures. The learned representations from ImageNeXt can serve as strong initialization across a spectrum of multi-modal scenarios, which we further demonstrate in downstream evaluations beyond RGB-D.
>
> We will make this broader perspective more explicit in the revised version. Thank you again for the helpful suggestions.
>
>
> **3.About the Table 2.**
>
> Due to space limitations, we did not include all combinations of RGB with a single supplementary modality in Table 2. Following your suggestion, we have conducted and now provide the complete results for RGB with a single additional modality in line 7-10 of the table below. It can be observed that pretraining RGB together with any additional modality consistently yields significant improvements in downstream tasks involving that modality.
>
>
> Table 2. Different modality settings within our ImageNeXt pretraining. Note that the pretraining
> duration is 100 epochs in this experiment. We mark the significantly dropped performance in **bold**.
> | Index | RGB | Depth | Event | LiDAR | Thermal | NYU V2 (RGB-D) | MFNet (RGB-T) | KITTI (RGB-L) | EventScape (RGB-E) | EventScape (RGB-D-E) |
> |-------|-----|-------|-------|--------|---------|----------------|----------------|----------------|---------------------|------------------------|
> | 1     | ✓   | ✓     | ✓     | ✓      | ✓       | 54.3           | 57.6           | 64.6           | 61.8                | 63.8                   |
> | 2     | ✓   |    |    ✓     | ✓      | ✓       | **52.2**       | 57.5           | 64.6           | 61.6  | **61.9**               |
> | 3     | ✓   | ✓     |       |    ✓     | ✓       | 54.2           | 57.6           | 64.5           | **60.5**               | **62.9**                   |
> | 4     | ✓   | ✓     |    ✓     |    |  ✓    | 54.3           | 57.7           | **61.2**       | 61.9                | 63.7                   |
> | 5     | ✓   | ✓     |   ✓      |    ✓     |      | 54.3           | **56.4**       | 64.8           | 62.1                | 63.8                   |
> | 6     | ✓   | ✓     | ✓     |       |         | 54.4           | **56.4**       | **61.4**       | 62.1                | 64.0                   |
> | 7     | ✓   |     |       |    | ✓         | **51.8** | 57.7     | **61.1**       | **60.4**            | **60.8**               |
> | 8     | ✓   |     |       |  ✓   |         | **52.0**  | **55.9**       | 64.8       | **60.7**            | **61.1**               |
> | 9     | ✓   |   |    ✓      |    |         | **51.6**           | **56.3** | **61.0**       |61.9  | **62.4**               |
> | 10     | ✓   | ✓     |       |    |         | 54.6           | **56.2**       | **61.3**       | **60.5**            | **63.1**               |
> | 11     | ✓   |   |       |       |         | **50.9**       | **55.6**       | **60.1**       | **58.7**            | **59.7**               |
>
> We will update the Table 2 of the main paper to incorporate these more comprehensive experiment results. We appreciate your suggestion, which has helped us improve the completeness and transparency of our experimental evaluation.

---

> > ### Comment · Reviewer_HYgf · 2025-08-06
> >
> > Thanks for the authors response, which solves my concern. Hoping the public of dataset.

---

> > > ### Author Response · Authors · 2025-08-06
> > >
> > > We sincerely appreciate the reviewer's time and effort in reviewing this paper. The dataset, codes, and checkpoints are well-organized and will be released once the paper is made public.

---

> ### Comment · Area_Chair_X6ma · 2025-08-05
>
> Dear Reviewer,
>
> Thank you for your dedicated efforts in reviewing this paper. We are currently in the reviewer-author discussion phase, but we have not yet seen your engagement.
>
> This year's Responsible Reviewing initiative requires all reviewers to communicate with authors during this period, emphasizing that ghosting is not acceptable. We kindly ask that you reply and engage with the authors. Please note that participation in discussions with authors is mandatory before submitting "Mandatory Acknowledgement," as submitting it without any engagement is not permitted in this review cycle.
>
> Best,
> AC

---

### Official Review · Reviewer_A5r8 · 2025-06-30

**Clarity:** 4
**Significance:** 4
**Originality:** 4
**Rating:** 5
**Confidence:** 4

**Summary:**

The paper presents **OmniSegmentor**, a novel framework for multimodal semantic segmentation. The proposed framework comprises two main components: (i) a new large-scale dataset, **ImageNeXt**, which includes RGB images along with four additional modalities, thermal, depth, LiDAR, and event data; and (ii) an elegant training method that enables efficient learning across multiple modalities and facilitates easy fine-tuning for downstream tasks.

The proposed method achieves state-of-the-art performance on several multimodal semantic segmentation benchmarks, including **NYU Depth v2**, **EventScape**, **MFNet**, **DeLiVER**, **SUN RGB-D**, and **KITTI-360**.

**Questions:**

See the questions in the Strengths And Weaknesses section.

**Ethical Concerns:**

["NO or VERY MINOR ethics concerns only"]

**Final Justification:**

After the rebuttal, all of my initial concerns were successfully addressed, so I chose to maintain my original positive rating.

**Limitations:**

yes

**Quality:**

4

**Strengths And Weaknesses:**

**Strengths**

* The paper introduces ImageNeXt, the first large-scale multimodal dataset that includes thermal, depth, LiDAR, and event data, alongside RGB images sourced from ImageNet. The dataset is constructed using synthetic data generated through multiple techniques, enabling the authors to extract all required modalities from RGB images.

* A novel pretrain-and-finetune pipeline is proposed, offering an efficient pretraining strategy combined with flexible fine-tuning. Specifically, to maximize efficiency during pretraining, the model is fed only the RGB image at each iteration, along with a randomly selected supplemental modality. This strategy significantly reduces training time, by more than a factor of two, compared to training on all modalities simultaneously, while maintaining the same number of parameters and FLOPs as RGB-only training.

* The proposed pipeline enables straightforward fine-tuning on downstream tasks by simply loading the pretrained weights and appending lightweight modules to capture the specific characteristics of each modality.

* OmniSegmentor achieves state-of-the-art results across all evaluated multimodal semantic segmentation benchmarks.

* The paper is well-written and easy to follow. The authors provide extensive experimental results to thoroughly validate the proposed approach.

---

**Weaknesses**

* **Concern about modality selection randomness.** While not a critical flaw, this point could benefit from further clarification. During pretraining, a supplemental modality is randomly selected at each iteration. Have you considered alternative, more deterministic selection strategies, such as a round-robin schedule (e.g., depth → thermal → LiDAR → event → [repeat])? Exploring such strategies might further enhance the robustness and effectiveness of the proposed pretraining approach.

---

> ### Author Rebuttal · Authors · 2025-07-30
>
> Thanks for your effort for reviewing our paper and giving some kind suggestions. We are happy to see your positive comments on writing, experiments and method. We hope the following responses could solve your concerns.
>
> **About modality selection randomness.**
> Thank you for the valuable comment regarding the modality selection strategy during pretraining.
> The ImageNeXt pretraining involves randomly sampling one supplementary modality at each iteration.
> Following your suggestion, we conduct experiments with a more deterministic round-robin schedule (e.g., depth → thermal → LiDAR → event → ...), and compare its effectiveness with our random selection strategy.
> As shown in the table below, we observe that they have similar performance across all evaluated multi-modal tasks. The results indicate that our pretraining method is robust to the choice of modality scheduling.
>
> | Sample Manner | NYU V2 (RGB-D) | MFNet (RGB-T) | KITTI (RGB-L) | EventScape (RGB-E) | EventScape (RGB-D-E) |
> |----|---------|----------------|----------------|----------------|---------------------|
> |  random sampling  | 57.6           | 60.6           | 69.2           | 65.0                | 67.6                   |
> | deterministic round-robin | 57.8       | 60.5           | 68.8           | 65.5            | 67.4               |
>
>
> We hypothesize this robustness stems from the fact that such scheduling differences effectively correspond to variations in the gradient update order, which become less significant as training iterations accumulate. In other words, given sufficient training iterations, both random sampling and fixed round-robin schedules tend to converge to similar solutions in practice.
>
> We sincerely thank the reviewer for this constructive suggestion, which provides additional insights and makes this study more in-depth.

---

> > ### Comment · Reviewer_A5r8 · 2025-08-06
> > **Thanks for the rebuttal**
> >
> > Dear Authors,
> > Thank you for addressing my concern regarding modality selection. I found the results quite interesting: it's fascinating that the deterministic approach performs comparably to random sampling. I believe this analysis would be a valuable addition to the final version of the paper, at least in the supplementary materials.
> >
> > Thank you again for your efforts.
> >
> > Best regards

---

> > > ### Author Response · Authors · 2025-08-06
> > >
> > > Thanks for the constructive feedback. We will incorporate the experiments and detailed analysis regarding the modality selection strategy into the paper. We greatly appreciate your suggestion, as it adds further insights and depth to our work.

---

> ### Comment · Area_Chair_X6ma · 2025-08-05
>
> Dear Reviewer,
>
> Thank you for your dedicated efforts in reviewing this paper. We are currently in the reviewer-author discussion phase, but we have not yet seen your engagement.
>
> This year's Responsible Reviewing initiative requires all reviewers to communicate with authors during this period, emphasizing that ghosting is not acceptable. We kindly ask that you reply and engage with the authors. Please note that participation in discussions with authors is mandatory before submitting "Mandatory Acknowledgement," as submitting it without any engagement is not permitted in this review cycle.
>
> Best,
> AC

---

### Author Response · Authors · 2025-08-05
**Rebuttal Summary and Response to Reviewer Comments**

Dear Reviewers and AC:

We sincerely appreciate your valuable time and constructive comments.
We are pleased that our work was recognized as "the first large-scale multimodal dataset including thermal, depth, LiDAR, and event data alongside RGB images” and “an elegant and efficient pretrain-and-finetune pipeline" (Reviewer A5r8), "a framework that achieves good results on multimodal semantic segmentation datasets" (Reviewer HYgf), "an efficient pretraining strategy outperforming naive multi-modality training" (Reviewer Eebt), and "a flexible, user-friendly framework achieving state-of-the-art results on various benchmarks" (Reviewer TR9e).
Several common and important concerns were raised, which we have addressed in detail:

* Performance gains on RGB-D benchmarks: While improvements over DFormer on RGB-D tasks appear modest, DFormer is highly specialized for RGB-D. Our method is more general and modality-agnostic, yet still delivers consistent gains while transferring effectively to a broader range of modalities and architectures.

* Synthetic data and real-world generalization: Synthetic data is used solely for pretraining, with fine-tuning on real-world datasets ensuring adaptation. Experiments across diverse indoor and outdoor benchmarks confirm that our strategy consistently improves real-world performance despite synthetic pretraining data.

We also addressed several reviewer-specific points, such as:

* Modality selection strategy (Reviewer A5r8): We compared our original random sampling with a deterministic round-robin schedule and found similar performance across all tasks, demonstrating the robustness of our pretraining approach to scheduling choices.

* Completeness of modality combination experiments (Reviewer HYgf): We extended Table 2 to include all RGB + single-modality settings, showing that pretraining with any additional modality consistently improves tasks involving that modality.

We believe we have thoroughly addressed all reviewer comments and have further clarified the scope and significance of our contributions. We welcome any additional feedback or questions and look forward to continued discussion during the rebuttal phase.

---

### Note · Authors · 2025-08-13

We sincerely thank all reviewers for their constructive feedback and active engagement throughout the review process. The discussions have helped us clarify our contributions, refine the presentation, and strengthen the experimental analysis of this paper.

The review phase highlighted several key strengths of our work: (1) OmniSegmentor introduces ImageNeXt, the first large-scale multi-modal dataset covering five modalities: RGB, depth, thermal, LiDAR, and event. It also provides a flexible, efficient pretrain-and-finetune pipeline applicable to arbitrary modality combinations; (2) our unified multi-modal pretraining strategy is robust to modality scheduling choices, as confirmed by additional “deterministic vs. random sampling” experiments; (3) the framework achieves state-of-the-art performance across diverse real-world benchmarks while maintaining competitive efficiency; and (4) the method demonstrates strong transferability and scalability, including to unseen modalities such as polarization.

We appreciate the recognition from Reviewer A5r8 for the effectiveness and robustness of our modality selection strategy, from Reviewer HYgf for the completeness of the updated modality combination experiments, from Reviewer Eebt for confirming the practical value and efficiency of our pretraining approach, and from Reviewer TR9e for emphasizing the framework’s flexibility, reproducibility, and broad applicability. We are encouraged that concerns raised during the review were addressed to the reviewers’ satisfaction, leading to strengthened support for acceptance.

We will incorporate all additional experiments, clarifications, and corrections into the revision, ensuring the paper fully reflects constructive discussions. We are grateful for the reviewers’ and AC’s efforts, which have helped us make this paper more solid.

---

### Decision · Program_Chairs · 2025-09-17

**Decision:**

Accept (poster)

**Comment:**

The reviewers were in unanimous agreement regarding the acceptance of this paper in their initial reviews. The author's rebuttal effectively addressed all remaining concerns, maintaining the positive consensus. Overall, the paper is well above the NeurIPS standard. Therefore, the AC recommends acceptance.